# *Mycobacterium tuberculosis* Central Metabolism Is Key Regulator of Macrophage Pyroptosis and Host Immunity

**DOI:** 10.3390/pathogens12091109

**Published:** 2023-08-30

**Authors:** Michelle E. Maxson, Lahari Das, Michael F. Goldberg, Steven A. Porcelli, John Chan, William R. Jacobs

**Affiliations:** 1Program in Cell Biology, The Hospital for Sick Children, Toronto, ON M5G 0A4, Canada; michelle.maxson@sickkids.ca; 2Department of Microbiology and Immunology, Albert Einstein College of Medicine, Bronx, NY 10461, USA; lahari.das@einsteinmed.edu (L.D.); steven.porcelli@einsteinmed.edu (S.A.P.); 3BostonGene, 100 Beaver St., Waltham, MA 02453, USA; michael.goldberg@bostongene.com; 4Department of Medicine, New Jersey Medical School, 205 South Orange Avenue, Newark, NJ 07103, USA; jc2864@njms.rutgers.edu

**Keywords:** pyroptosis, *Mycobacterium*, caspase-1, Rv3727, redox balance

## Abstract

Metabolic dysregulation in *Mycobacterium tuberculosis* results in increased macrophage apoptosis or pyroptosis. However, mechanistic links between *Mycobacterium* virulence and bacterial metabolic plasticity remain ill defined. In this study, we screened random transposon insertions of *M. bovis* BCG to identify mutants that induce pyroptotic death of the infected macrophage. Analysis of the transposon insertion sites identified a panel of *fdr* (*f*unctioning *d*eath *r*epressor) genes, which were shown in some cases to encode functions central to *Mycobacterium* metabolism. In-depth studies of one *fdr* gene, *fdr8* (BCG3787/Rv3727), demonstrated its important role in the maintenance of *M. tuberculosis* and *M. bovis* BCG redox balance in reductive stress conditions in the host. Our studies expand the subset of known *Mycobacterium* genes linking bacterial metabolic plasticity to virulence and also reveal that the broad induction of pyroptosis by an intracellular bacterial pathogen is linked to enhanced cellular immunity in vivo.

## 1. Introduction

*Mycobacterium tuberculosis* is a highly effective pathogen that has evolved numerous mechanisms to successfully invade, replicate, and persist in the macrophage (MΦ) of humans and mammals. Persistence in MΦ in vivo contributes to the chronicity of tuberculosis (TB), the infectious disease caused by members of the *M. tuberculosis* complex. To effectively persist, intracellular mycobacteria must not only evade innate pathogen recognition mechanisms and antimicrobial pathways but also metabolically adapt to the MΦ intracellular environment. The successful synthesis of these strategies allows for the tubercle bacilli to persist and multiply within the host. 

In vivo, innate immune cells such as MΦ are major cellular reservoirs for persistent intracellular mycobacteria [1,2]. Following infection of the host by infectious aerosol, lung-resident MΦ are amongst the first cells to encounter the bacterium. Within MΦ, mycobacteria initiate global changes in gene expression, upregulating virulence factors and pathways for fatty acid degradation, gluconeogenesis, and iron acquisition [3,4,5,6,7]. For intracellular pathogens such as mycobacteria, pathogenicity is correlated to metabolic plasticity in vivo and the adaptation to host lipid carbon sources [8,9,10,11,12]. These observations highlight the importance of bacterial metabolism to the host–pathogen interaction, although the mechanisms for this linkage remain to be clarified.

Virulent mycobacteria inhibit several MΦ cell death pathways, and in some cases, this has been linked to effects of mycobacterial infection on metabolic homeostasis [13,14,15,16,17,18,19,20,21,22,23]. The derepression of apoptosis or pyroptosis by precise deletions of mycobacterial genes (∆*secA2*, ∆*nuoG*, ∆*zmp1*, and ∆*ptpB*) results in attenuation of virulence [13,15,17,24,25,26,27]. Surprisingly, the products of these mycobacterial genes are mainly involved in bacterial respiration and intermediate metabolism [28]. In addition, a partial screen of 5000 *M. tuberculosis* transposon mutants for effects on growth in human myeloid lineage cells identified 27 genes that affect MΦ apoptosis and pyroptosis, 15 of which are involved in intermediate or lipid metabolism [22,23]. These findings emphasize the connections between host cell death and mycobacterial intracellular metabolic adaptation. 

In the current study, we used a global genetic analysis to investigate a well-known mycobacterial innate immune evasion tactic, the suppression of MΦ cell death, as a means to broadly interrogate requirements for pathogen virulence and host immunity. By screening an *M. bovis* BCG transposon (Tn) mutant library for mutants that caused death of infected human MΦ, we identified a panel of *fdr* (*f*unctioning *d*eath *r*epressor) mutants that induced robust pyroptotic cell death in human MΦ. A number of these mutations mapped to genes central to *Mycobacterium* metabolism, supporting a link between mycobacterial metabolic homeostasis and the host MΦ response to infection. One *fdr* gene, *fdr8* (BCG3787/Rv3727), encoded a hypothetical oxidoreductase/phytoene desaturase important for the maintenance of *M. tuberculosis* and *M. bovis* BCG redox balance in reductive stress conditions. In addition, *fdr*-induced MΦ pyroptosis resulted in enhanced *M. tuberculosis*-specific CD4 and CD8 T-cell immunity in vivo. This work presents data connecting bacterial pyroptosis induction and in vivo protective immunity. Our studies underscore the intimate connection between *M. tuberculosis* intracellular metabolic plasticity and innate immune evasion, an essential aspect of mycobacterial pathogenesis in the human host.

## 2. Materials and Methods

### 2.1. THP-1 and Primary Human MΦ Infection with Mycobacteria Strains

After differentiation into adherent MΦ by overnight treatment in 50 nM phorbol 12-myristate 13-acetate (PMA, Sigma-Aldrich, St. Louis, MO, USA), the medium was changed to complete RPMI medium (see Appendix A) containing 5% non-heat inactivated human serum. Adherent THP-1 cells were infected an MOI of 5 bacteria to 1 MΦ (5:1) for 3 h. After infection, infected THP-1 monolayers were washed 3 times with 1X PBS to remove extracellular bacteria and left in complete RPMI media at 37 °C and 5% CO_2_ until analysis. Most analyses were carried out 3–5 days post infection, as indicated. For infections with human primary cells, the same infection protocol was followed, with the exception that pre- and post-infection monolayers were maintained in RPMI media containing 10% AB human serum.

### 2.2. Flow Cytometric Sub-G_1_ Cell Cycle Analysis of Infected THP-1 Cells

Procedure was adapted from Riccardi et al. 2006 [29]. Briefly, 3 days post infection, the media from mycobacteria-infected THP-1 monolayers were removed from micotiter plates using 0.1% trypsin-EDTA (Invitrogen, Waltham, MA, USA) and mechanical pipetting. THP-1 cells were washed in 1× PBS, and fixed in ice-cold 70% ethanol at −20 °C. The following day, fixed cells were washed in 1× PBS and resuspended in propidium iodide staining solution (0.1% Triton X-100, 20 µg·mL^−1^ propidium iodide, 200 µg·mL^−1^ RNase A). Cells were incubated 30 min at room temperature in the dark. Cells were then subjected to flow cytometric cell cycle analysis using a FACSCalibur cytometer (BD Biosciences, Franklin Lakes, NJ, USA). Data were further analyzed with the FlowJo software (v.9, Tree Star, Inc., San Carlos, CA, USA). A hypodiploid, i.e., sub-G_1_, population in the FL2-H channel was indicative of DNA fragmentation that accompanies apoptotic or pyroptotic cell death. 

### 2.3. Detection of Active Caspase-1 and Caspase-3/7

FAM-FLICA in vitro Caspase Detection Kits for caspase-1 and caspase-3 (Immunochemistry technologies) were used to detect active caspases in infected THP-1 cells by microscopy. Briefly, 4 × 10^5^ THP-1 cells were seeded onto 12 mm glass coverslips in 24-well plates following differentiation procedures (Appendix A). Infections with various mycobacteria strains were performed as described (Appendix A), and 3 days post infection, coverslips were stained with FAM-YVAD-fmk or FAM-DEVD-fmk to detect active caspase-1 or caspase-3/7, respectively, and fixed according to the manufacturer’s instructions for adhered cells. Stained monolayers were visualized under 200× magnification and the green channel using an Olympus IX 70 inverted microscope, equipped with a 100 W Hg light source, a lambda filter wheel containing 61002 DAPI/FITC/Texas Red cube (61002 exciter and 61002bs dichroic), and blue (BP 330-385), cyan (S403/12), green (S492/18), and red (S572/23) fluorescence channels. Images were captured using an ORCA-II Hamamatsu camera controlled by MetaMorph software (version 3.5, Molecular Devices). Image processing and analyses were done using Photoshop CS5 (Adobe, San Jose, CA, USA) or ImageJ (version 2, Wayne Rasband, U.S. National Institute of Health) software. For each sample, cells were counted from three different fields, with 100 cells counted per field. 

### 2.4. Caspase and Inflammasome Inhibitor Studies

After infection with various mycobacteria strains, THP-1 monolayers were placed in complete RPMI media containing 50 µM Z-YVAD-FMK (Millipore, Burlington, MA, USA) or Z-DEVD-FMK (Millipore) to inhibit caspase-1 or caspase-3, respectively. Three days post infection, infected THP-1 cells were subjected to sub-G_1_ analysis and flow cytometry, as described above. Similarly, infected THP-1 monolayers were treated with 50 µM glyburide (Millipore) to inhibit the NALP3 inflammasome and subsequently analyzed by flow cytometric sub-G_1_ analysis.

### 2.5. Multiplex Cytokine Analysis of Infected THP-1 Cell Supernatants

Cytokine analysis was performed on supernatants from THP-1 cells infected with various mycobacteria strains. Infection of THP-1 cells was carried out as described above, and 4 days post infection, supernatant samples were collected for analysis. MSD MULTI-ARRAY human T_H_1/T_H_2 10-plex kit (Meso Scale Diagnostics LLC, Rockville, MD, USA) was used to analyze 25 µL of supernatant containing secreted human cytokines, according to the manufacturer’s instructions. The human T_H_1/T_H_2 10-plex kit was used to detect secreted IFN-γ, IL-1β, IL-2, IL-4, IL-5, IL-8, IL-10, IL-12p70, IL-13, and TNF-α. MULTI-ARRAY plates were read using the MSD SECTOR Imager 2400, following the manufacturer’s instructions.

### 2.6. Immunizations and Enzyme-Linked Immunosorbent Spot (ELISPOT) Assay

Mycobacteria strains were grown to mid-log phase and resuspended in 1X PBS containing 0.05% (*v*/*v*) tyloxapol. A 1 × 10^6^ cfu total dose was subcutaneously injected into C57BL/6 mice in both leg flanks, with 3 mice per group. At 21 days post immunization, animals were sacrificed and spleens isolated. Splenic T cells were separated using the Pan T-cell isolation kit and AutoMACS cell separator (Miltenyi Biotec, Bergisch Gladbach, Germany), according to the manufacturer’s instructions. Purified T cells (1.5 × 10^5^ cells per well) and naïve splenocytes (5 × 10^5^ cells per well) were resuspended in complete RPMI media and seeded into MultiScreen_HTS_ plates pre-coated with 10 µg·mL^−1^ IFN-γ capture antibody (rat α-mouse IFN-γ antibody (clone R4-6A2, BD Biosciences, Franklin Lakes, NJ, USA)). Then, 5 µg·mL^−1^ of peptide specific to mycobacterial CD4 (Peptide-25 (H2N-FQDAYNAAGGHNAVF-OH, >85% purity, New England Peptide)) or CD8 (TB10.3/4 (H2N-QIMYNYPAM-OH, >85% purity, New England Peptide)) T-cell responses were added to ELISPOT plates. Plates were incubated at 37 °C, 5% CO_2_ for 16 h. Plates were then developed using 1 µg·mL^−1^ biotin rat α-mouse IFN-γ detection antibody (clone XMG1.2, BD Biosciences, Franklin Lakes, NJ, USA) overnight (approximately 20 h), 1:800 diluted streptavidin-alkaline phosphatase (Invitrogen, Waltham, MA, USA), and BCIP substrate solution (1 tablet 5-bromo-4-chloro-3-indolyl-phosphate (BCIP)/nitroblue tetrazolium tablets (Sigma-Aldrich, St. Louis, MO, USA) dissolved in 10 mL H_2_O). The frequency (spot forming units, SFU) of lymphocytes secreting IFN-γ in response to *Mycobacterium* peptides was determined using an automated ELISPOT reader (Autoimmun Diagnostika GmpH, Straßberg, Germany).

### 2.7. Mycobacterium Spotted Microarray Analysis

Microarray analysis was performed on RNA isolated from *Mycobacterium* grown in standard 7H9 broth conditions or isolated from THP-1 cells, as indicated. Spotted *M. tuberculosis* H37Rv DNA microarrays were obtained through the U.S. National Institute of Allergy and Infectious Diseases-sponsored Pathogen Functional Genomics Resource Center (PFGRC) at the J. Craig Venter Institute (JCVI, La Jolla, CA, USA). cDNA probes were prepared from mycobacteria’s 4 µg RNA following the PFGRC microarray laboratory protocol SOP # M007. Cy3- and Cy5-labeled cDNA probes were hybridized according to PFGRC protocol SOP # M008 to 70-mer oligo DNA microarrays representing the complete *M. tuberculosis* genome (*M. tuberculosis* v. 4 microarray slides). Three biological replicate microarrays were carried out per sample. Slides were scanned on a GenePix 4000A scanner (Molecular Devices, San Jose, CA, USA). Images were processed with the TM4 software suite37 (JCVI). TIGR Spotfinder was used to grid and quantitate spots. TIGR MIDAS was used for Lowess normalization, S.D. regularization, and in-slide replicate analysis, with all quality-control flags on and one bad-channel-tolerance policy set to “generous”. Results were analyzed in TIGR MeV with significance analysis of microarrays (SAM) and considered significant at *q* < 0.05. 

### 2.8. Infected-THP-1 Cell Affymetrix Microarray Analysis

Microarray analysis was performed on RNA isolated from mycobacteria-infected THP-1 cells, as described above. First, 700 ng total RNA was further processed by the Albert Einstein College of Medicine Microarray Facility using the standard Affymetrix pipeline and hybridized to a Human Gene 1.0 ST GeneChip array (Affymetrix, Santa Clara, CA, USA). The raw data CEL files, provided by the Microarray Facility, were normalized by RMA methods in GeneSpring GX software (Affymetrix). The log_2_-transformed signal intensities were averaged for biological triplicates and the mean value used to compute the fold expression change. Results were analyzed in TIGR MeV with significance analysis of microarrays (SAM) and considered significant at *q* < 0.05. 

Significant data were analyzed through the use of IPA (Ingenuity Systems, www.ingenuity.com, accessed on 12 December 2012). Functional analysis identified the biological functions from the Ingenuity Knowledge Base most significant to the dataset and the dataset molecules associated with these functions. Right-tailed Fisher’s exact test was used to calculate *p*-value, determining the probability that each biological function assigned to the dataset was due to chance alone.

### 2.9. Mycobacteria Oxidative and Reductive Stress Exposure Experiments

Strains were grown to mid-log phase, with an approximate OD_600nm_ = 0.5–1 in standard media. For oxidative stress, 1.5 × 10^7^ cfu of bacteria were incubated in 7H9-based media containing 0.5% BSA, 0.2% glycerol, and 0.05% tyloxapol with 0, 0.01%, 0.1%, or 1% H_2_O_2_. For reductive stress, 1.5 × 10^7^ cfu of bacteria were incubated in the same 7H9-based media containing 0, 0.5 mM, 5 mM, or 50 mM DTT. Samples were incubated at 37 °C for up to 5 days, and samples were spotted for cfu every 24 h.

### 2.10. M. bovis BCG and M. tuberculosis Growth Curves

Ten milliliters of *Mycobacterium* seed cultures were grown to mid-log phase in standard media (see Appendix A). Bacteria were resuspended in 7H9 base media (no carbon source) and used to inoculate 10 mL cultures to an OD_600nm_ of 0.05. *M. bovis* BCG-derived strains were grown in 7H9 broth containing 100 mM MOPS pH 6.6, 0.05% (*v*/*v*) tyloxapol, and a single carbon source at the following concentrations: 10 mM glucose, 20 mM glycerol, 30 mM acetate, 20 mM propionic acid, 15 mM butyric acid, 12 mM valeric acid, 10 mM caproic acid, 250 µM cholesterol, and 25 µM 1,2-dipalmitoyl-*sn*-glycero-3-phosphocholine (DpPC). *M. tuberculosis* H37Rv-derived strains were grown in 7H9 broth containing 100 mM MOPS pH 6.6, 0.5% (*w*/*v*) BSA, 0.085% (*w*/*v*) NaCl, 0.05% (*v*/*v*) tyloxapol, and a single carbon source at the following concentrations (*w*/*v*): 0.2% glucose, 0.2% glycerol, 0.06% acetate, 0.06% propionic acid, 0.06% butyric acid, 0.06% valeric acid, and 0.016% caproic acid. Cultures were incubated at 37 °C and monitored for 4 weeks by OD_600nm_. 

### 2.11. Statistical Analyses

GraphPad Prism 5 and Microsoft Excel (version 2011) were used for statistical analyses. The F-test was used to determine equal variance between experimental data and control (*p* > 0.05). The unpaired, two-tailed *t*-test was used to evaluate significance relative to control, as indicated in the text. Significance is displayed according to the Michelin Guide scale: * *p* < 0.05; ** *p* < 0.01; *** *p* < 0.001; ns: not significant. 

For growth curves, Graphpad Prism 5 software was used for non-linear regression Logistic curve fit analysis, using the following formulas: y = y_max_ ∗ y_0_/[(y_max_ − y_0_) ∗ exp(−k ∗ x) + y_0_)]
where y_0_ is the OD_600nm_ at time zero, and y_max_ is OD_600nm_ at plateau.
k = growth rate = ln 2/doubling time.

The typical growth rate for a bacterial strain with a 24 h doubling time is 0.029 h^−1^.

## 3. Results

### 3.1. Identification of Mycobacterium Mutants That Promote MΦ Cell Death

The strain chosen for Tn mutagenesis was mc^2^6455, a recombinant *M. bovis* BCG Danish 1331 strain (see Appendix A). *M. bovis* BCG is genetically 99.95% identical to *M. tuberculosis* and preserves many mycobacterial immune evasion mechanisms, including the suppression of host cell death [17,30]. Strain mc^2^6455 expresses the *Clostridium perfringens* perfringolysin O [31,32], predicted to mediate bacterial endosomal escape in a manner similar to *M. tuberculosis* RD1-mediated phagosomal escape [33,34,35]. Therefore, mc^2^6455 is expected to partially mimic the phenotype of virulent RD1^+^ *M. tuberculosis* and *M. bovis* in intracellular lifestyle. 

Tn mutagenesis was performed on mc^2^6455, and 5000 Tn mutants were screened by infection of human monocytoid THP-1 cells to identify mutants defective in the suppression of MΦ cell death. Candidate Tn mutants were selected on the basis of increased numbers of dead THP-1 cells based on microscopy of cultures stained with the fluorescent dye HO258. Initial screening identified 17 candidate mutants that had substantially lost the ability of the parental mc^2^6455 to suppress MΦ cell death. These mutants were subjected to secondary screening using an alternate cell death detection method in which infected MΦ were analyzed by flow cytometry to quantitate the proportion of cells showing substantial loss of genomic DNA due to DNA fragmentation [29]. These results confirmed the robust induction of MΦ cell death by 9 of the Tn mutants (Figure 1A), which we designated as *fdr* (*f*unctioning *d*eath *r*epressor) mutants. The percent cell death in THP-1 cells following infection with *fdr* mutants was similar to that observed in *M. kansasii*-infected control cultures. In these experiments, treatment of THP-1 cells with 1 µM staurosporine (an inducer of apoptosis) or 20 µM nigericin (an inducer of necrosis) were also included as controls. 

To prove that the *fdr* genes were responsible for the induction of MΦ cell death, complementation analysis was performed. Specific complementing cosmids were transformed into corresponding *fdr* mutants, and these strains were used to infect THP-1 cells. The results indicated that the introduction of the appropriate cosmids clearly complemented the cell death phenotype (Figure 1B). Therefore, these *fdr* mutant phenotypes were specific to an *fdr* gene knockout.

The Tn insertion sites for these five *fdr* mutants were mapped to the *M. bovis* BCG genome (Table 1). Of note, one ORF was identified by two independent Tn hits (BCG3787, in *fdr8* and *fdr15*); therefore, *fdr8* was used for all subsequent experiments. The *fdr* ORF functional classifications revealed that four out of the five ORFs identified (*fdr2*, *fdr4*, *fdr8/15*, and *fdr16*) were involved in intermediate metabolism or lipid metabolism. This abundance of metabolic genes suggests a major role for the metabolic state of intracellular bacteria in the induction of cell death in infected MΦ and in the inhibition of this process by pathogenic mycobacteria.

### 3.2. Mechanism of Cell Death in MΦ Infected with fdr Mutants

Preliminary cell death analyses further defined the MΦ cell death induced by the *fdr* mutants. *fdr*-infected THP-1 monolayers were subjected to terminal deoxynucleotidyl transferase dUTP nick-end labeling (TUNEL staining). Microscopy analysis showed that all the *fdr*-infected THP-1 cells showed increased nuclear TUNEL staining (Appendix A). In addition, *M. kansasii*-infected and staurosporine-treated cells showed robust TUNEL staining. These data corroborated the results of sub-G_1_ analysis (Figure 1A). However, TUNEL staining can be seen in cells undergoing DNA fragmentation associated with apoptosis or pyroptosis [23,36].

In addition, infected THP-1 monolayers were stained for pan-caspase activation. Microscopic analysis showed that all *fdr*-infected THP-1 cells were caspase-positive (Appendix A). *M. kansasii* controls were also caspase-positive. Between the cell death modes that result in HO258 positivity, apoptosis and pyroptosis are caspase-positive, while necrosis is caspase-negative [37]. Taken together, these data suggested that the cell death induced in THP-1 cells by *fdr* mutants is not necrotic but more representative of apoptosis or pyroptosis. 

### 3.3. fdr Mutants Induced Caspase-1, NALP3-, and ROS-Dependent Pyroptosis

The *fdr*-mutants induced cell death in THP-1 cells sharing features with apoptosis and pyroptosis. Therefore, to distinguish between apoptosis and pyroptosis in *fdr*-infected MΦ, the involvement of caspase-3 or caspase-1 in *fdr*-induced cell death was assessed [38,39]. Initially, *fdr*-infected THP-1 cells were analyzed for the presence of active caspase-3/7 or caspase-1. In *fdr*-infected THP-1 cells, caspase-3/7 and -1 were activated equally, implying both were induced in the same infected cells (Figure 2A,C). The *M. kansasii* control showed active caspase-3 and caspase-1, consistent with previous reports [14,40,41]. Staurosporine induced active caspase-1; however, a study by Danelishvili et al. (2011) reported caspase-1 induction after staurosporine treatment [23]. Active caspase-3/7 and caspase-1 are both seen during pyroptosis [42,43,44]; however, apoptosis is A caspase-3-dependent cell death, while pyroptosis is caspase-1-dependent. Therefore, these data suggested the cell death induced by the *fdr* mutants was pyroptosis.

To further demonstrate that *fdr*-induced cell death was pyroptosis, inhibitor studies were performed. Caspase-1 is solely required for pyroptotic cell death [38,42,44,45,46]. Infected THP-1 cells’ monolayers were treated with 50 µM caspase-3/7 or caspase-1 inhibitor (Figure 2B,D). Cell death analyses showed that while caspase-3/7 inhibitor had no effect on *fdr*-induced THP-1 cell death (Figure 2B), caspase-1 inhibitor significantly decreased *fdr*-induced cell death to control levels (Figure 2D). Therefore, caspase-1 was required for *fdr*-induced cell death, consistent with similar pyroptosis studies using the avirulent *M. kansasii* [41]. Taken together, these results indicate that the cell death induced by *fdr* mutants in THP-1 cells was not apoptotic but pyroptotic in nature.

Pyroptosis involves the activation of caspase-1 by the recognition of intracellular danger signals by Nod-like-receptors, culminating in the secretion of IL-1β and caspase-1-dependent cell death [37,46,47]. Pyroptosis induced by *M. kansasii* involves the Nod-like-receptor NALP3 [41]. Therefore, the requirement for NALP3 inflammasome in pyroptosis induced by *fdr* mutants was tested by using a well-known inhibitor of NALP3, glyburide [48]. Cell death analyses confirmed that glyburide significantly decreased *fdr*-induced MΦ cell death to that of the BCG parent control (Figure 2E). This expected decrease was also seen in the *M. kansasii* control. 

Because reactive oxygen species (ROS) have been shown to be a danger signal mediating activation of the NALP3 inflammasome [49,50] and are involved in *M. kansasii*-induced pyroptosis [41], the effect of an ROS scavenger N-acetylcysteine (NAC) was tested. As a result, 20 mM NAC reduced pyroptosis induced by *fdr* mutants by approximately 50%, similar to the *M. kansasii*-infected MΦ (Figure 2F). Therefore, the pyroptosis seen in THP-1 cells infected with *fdr* mutants was dependent on the NALP3 inflammasome, with intracellular ROS contributing to the pyroptosis induction.

### 3.4. MΦ Infected with fdr Mutants Secreted IL-1β and Exhibited a Host-Protective Cytokine Profile 

The activation of caspase-1 in pyroptotic cells results in the processing of pro-IL-1β into mature IL-1β, which is secreted [51]. IL-1β can have autocrine and paracrine effects on local pro- and anti-inflammatory cytokine secretion. Therefore, THP-1 cells infected with *fdr* mutants were analyzed for secretion of T_H_1 and T_H_2 cytokines by MSD MULTI-ARRAY technology at 4 days post infection. *M. kansasii* and *M. tuberculosis* H37Ra-infected THP-1 cell supernatants were included as cell-death-positive controls [14]. 

In all cases, the cytokine profile of *fdr* mutant-infected THP-1 cells was similar to the cell death control infections (Figure 3). *fdr* mutant-infected cells showed significantly increased IL-1β, consistent with pyroptosis (Figure 3A). They also secreted significantly increased TNF-α, less IL-8, and increased IL-4 (Figure 3B–D). Unexpectedly, *fdr* mutant-infected THP-1 cells produced increased IFN-γ (Figure 3E), which is normally secreted primarily by T cells but has also been found to be secreted by human MΦ in the presence of IL-12 and IL-18. Here, IL-12 showed a tendency to be increased (Figure 3F), but IL-18 was not tested. Overall, *fdr* mutant-infected MΦ exhibited a pro-inflammatory cytokine profile that included IL-4 and IFN-γ, two cytokines important for the control of mycobacterial infection and inflammation in the host. 

Interestingly, the cell death and cytokine responses to infection with *fdr* mutants was correlated with an intracellular bacterial growth restriction prior to the appearance of cell death (Appendix A). For five mutants (*fdr2*, *fdr4*, *fdr8*, *fdr11*, and *fdr16*), the approximately half-log decrease in cfu at day 3 was statistically significant (between *p* < 0.01 and *p* < 0.05). The *fdr* strains had no significant growth differences 7H9 broth medium (Appendix A), indicating that these mutants did not have intrinsic growth defects. Together, these data suggest that the effects of *fdr* mutations on MΦ cell death and cytokine production were responsible for the enhanced control of intracellular burden.

### 3.5. Augmentation of CD4 and CD8 T-Cell Responses by Infection with fdr Mutants

As a pro-inflammatory, T_H_1-skewed immune response is considered optimal for the control of TB and the establishment of immunity to mycobacteria [52,53,54]. Pyroptosis has also been hypothesized to play an important role in bacterial immunity in vivo; however, direct evidence has been lacking [55]. Our results showing that pyroptotic *fdr* mutants induced a pro-inflammatory cytokine profile and have a growth defect in MΦ were consistent with the proposed role of pyroptosis in promoting host immunity. To further asses this in vivo and examine the effects of pyroptosis on adaptive immunity, the stimulation of IFN-γ producing T-cell responses in mice infected with *fdr* mutants was assessed by ELISPOT. BCG parent and *fdr* mutant bacteria were injected subcutaneously into C57BL/6 mice, and 21 days later, *M. tuberculosis*-specific CD4 and CD8 T-cell responses in the spleen were quantitated by IFN-γ ELISPOT (Figure 3G,H). BCG Danish strain 1331 was included for comparison to the BCG parent (mc^2^6455), and these showed similar results. Remarkably, the *fdr* mutants robustly enhanced both CD4 and CD8 responses to peptides from *M. tuberculosis* antigens Ag85B and TB10.3/4 (Figure 3G,F). The differences in responses between BCG parent and mutants were not due to differences in inoculum or bacterial burden at harvest since, despite the enhanced immunogenicity, the bacterial loads in the spleens at 3 weeks showed no significant differences between BCG parent and *fdr* strains (Appendix A). 

### 3.6. Microarray Analysis of fdr8 Intracellular Expression Patterns

Because the BCG3787 was identified twice in the Tn mutagenesis screen (*fdr8* and *fdr15*; Table 1), and these Tn insertions could be complemented with an appropriate cosmid (Figure 1B), this *fdr* locus was selected for more detailed analysis. To gain initial insight into the interaction of the *fdr8* mutant bacilli with host MΦ, we carried out global analyses by microarrays of both the bacterial and host cell transcriptomes. For intracellular bacterial gene expression analysis, mc^2^6455 (parental strain) and *fdr8* mutant strain were used to infect THP-1 cells, and bacterial RNA isolated 3 days post infection was used for microarray analysis. This revealed 294 genes differentially expressed at least 2-fold in *fdr8* mutant compared to the parental strain bacilli. Among these, 222 genes were over-represented (Appendix A), and 72 genes were under-represented (Appendix A) in the *fdr8* mutant. Amongst the over-represented genes was a large subset involved in mycobacterial fatty acid and protein biosynthesis (Figure 4, in red). 

Since it is hypothesized that lipid anabolism functions in redox balance to alleviate reductive stress [56,57], these data suggest that the *fdr8* mutant was undergoing reductive stress in MΦ. Consistent with this, eight genes involved in the mycobacterial response to stress from reactive oxygen species (ROS) were also over-represented (Figure 4, in red). Reductive stress can also paradoxically lead to production of ROS, resulting from the release of redox-active iron leading to oxidative stress [57,58]. Therefore, this implies the *fdr8* mutant experienced greater redox stress in MΦ and provides a possible explanation for the partial decrease in *fdr*-induced pyroptosis that was observed after NAC treatment (Figure 2). 

Also, there were over-represented genes in the *mce1* and ESX-5 regions, which are both predicted to have roles in *M. tuberculosis* pathogenesis. The *mce1* genes are expressed early in *M. tuberculosis* infection and induce pro-inflammatory cytokines [59]. The ESX-5 system is expressed intracellularly and induces caspase-1, secretion of IL-1β, and pyroptotic cell death in THP-1 MΦ [60]. Therefore, the over-representation of *mce1* and ESX-5 genes may contribute to the effects of *fdr8* mutant bacilli on MΦ cytokine and cell death responses. 

Amongst the 72 under-represented genes, many were conserved hypothetical proteins. However, consistent with the over-representation of redox stress-related genes, the under-represented genes included genes that increase bacterial iron acquisition and storage that could potentially promote redox damage (Figure 4, in blue) Of note, there were several uncharacterized, transcriptional regulators under-represented in the *fdr8* mutant, which could be involved in regulation of these redox genes. 

### 3.7. Microarray Analysis of fdr8-Infected MΦ Gene Expression

Gene expression profiles of THP-1 cells infected with *fdr8* mutant versus parental strains were also assessed by microarray, revealing 80 genes that were differentially expressed. To determine whether any biological pathways or functions were overrepresented within this gene set, ingenuity pathway analysis (IPA) was performed (Ingenuity Systems).

Overall, 67 genes were over-represented in *fdr8* mutant-infected THP-1 cells (Appendix A) and 13 genes under-represented (Appendix A) compared to wild-type infected MΦ. The expression data and IPA analyses support previous observations that pyroptosis and a pro-inflammatory innate response was induced in *fdr8* mutant-infected MΦ (Appendix A). IPA identified the cell-to-cell signaling and interaction, cellular movement, and immune cell trafficking functions to be most highly over-represented in this pattern of gene expression. (Appendix A). On the other hand, cell-mediated immune responses, hematological system development, lymphoid tissue structure and development, and cell-to-cell signaling and interaction functions were the most under-represented (Appendix A).

### 3.8. Sensitivity of fdr8-Null Mutants to Reductive Stress

In order to test the impact of the *fdr8* locus on redox stress, we tested the impact of both a classical oxidative stress inducer (0.1% hydrogen peroxide) and a reductive stress inducer (5 mM dithiothreitol, DTT) on the growth of mycobacteria in culture. For these experiments, we generated precise null deletions of this locus in the parental mc^2^6455 BCG strain as well as standard BCG-Danish and in an attenuated strain of *M. tuberculosis* (Appendix A; mc^2^6206-*M. tuberculosis* H37RV Δ*panCDΔleuCD*) [61]. Of note, all null mutants induced pyroptosis in THP-1 cells independent of strain background (Appendix A). In addition, this induction of cell death was seen in human primary MΦ (Appendix A). The *fdr8*-null mutants of BCG-Danish and mc^2^6206 were used in further studies. 

In redox stress experiments, *M. bovis* BCG and *M. tuberculosis* strains showed similar results (Figure 5). 

The *fdr8* nulls were significantly more sensitive to 5 mM DTT treatment (reductive stress) than the wild type (Figure 5A,B). In contrast, the *fdr8*-null bacteria did not show growth inhibition in the presence of 0.1% H_2_O_2_ (Figure 5C,D). Rather, there was a trend for increased cfu of the BCG-Danish mutant and a statistically significant increase in cfu for the *M. tuberculosis* mutant. This enhancement, while unexpected, may reflect the over-representation of several ROS scavenging genes observed in the intracellular transcriptome of *fdr8* mutants (Figure 4). In contrast, the reductive stress induced by DTT treatment may represent a different form of redox stress that the *fdr8*-null mutant cannot overcome.

### 3.9. Defect in the Utilization of Odd-Chain Fatty Acids by fdr8 Mutants 

Many of the identified *fdr* genes have metabolic roles, including *fdr8* (Table 1). These strains had a growth defect in MΦ (Appendix A). Therefore, it is possible that the inactivation of *fdr* genes may affect bacterial metabolism and growth on in vivo carbons sources. Moreover, the *fdr8*-null mutants exhibited a defect in reductive stress conditions (Figure 5), and reductive stress has been correlated with mycobacterial catabolism of host lipid carbon sources [57]. Therefore, we investigated the effect of *fdr8* inactivation on growth during a 4-week period in media containing a variety of individual carbon sources (glucose, glycerol, acetate, propionate, butyrate, valerate, and caproate). In the *M. bovis* BCG and *M. tuberculosis fdr8*-null mutants, there were strong growth defects when the odd-chain lipids propionate and valerate were used as sole carbon sources (Appendix A). This was reflected in decreased growth rates compared to wild-type parent strains (Table 2, decreased growth rate has been bolded). The BCG-Danish *fdr8* mutant also had small defects in glucose and glycerol, while the *M. tuberculosis* mutant did not. The reason for this disparity in glucose and glycerol defects is currently not known but may represent species-specific differences in carbon utilization. Regardless, the odd-chain lipid growth defects in both *M. bovis* BCG and *M. tuberculosis fdr8*-null mutants strongly suggested that the *fdr8* locus has a conserved role in mycobacterial catabolism or detoxification of host cell lipids.

## 4. Discussion

### 4.1. fdr Mutations Map to a Diverse Set of Genes Conserved in Virulent Mycobacterium

In this study, we identified that novel *fdr* mutants induce MΦ cell death. Unlike the cell-death-repressing genes of other pathogenic bacteria that directly modulate host cell death machinery (such as Yop proteins of *Yersinia*) [62], a number of *fdr* Tn insertions mapped to genes involved in metabolism, albeit with uncharacterized or hypothetical functions (Table 1). Nonetheless, the *fdr* mutants characterized exhibited normal growth kinetics in vitro. Their growth defects in MΦ suggests the metabolic pathways identified are important for mycobacterial intracellular growth. This is supported by the observed in vitro growth defect of *fdr8* on odd-chain fatty acids; however, the defects of other *fdr* mutants on in-vivo-relevant carbon sources remain to be explored. Most are implicated in *M. tuberculosis* pathogenicity. *fdr2* (Rv1831) is expressed in both murine and human MΦ and is required for survival in primary murine MΦ [5,63]. *fdr4* (*tcrX*) is implicated in the response of *Mycobacterium* to iron starvation in the host [64] and is required [63] for survival in primary mouse MΦ. *fdr8* (Rv3727) is required for *M. tuberculosis* survival in non-human primates [65]. *fdr16* (*arcA*) is also required for *M. tuberculosis* survival in non-human primates [65]. These *fdr* genes indicate that mycobacterial metabolic balance in the host is paramount to the mycobacterial–MΦ interaction. However, it is possible that with reduced stringencies, future genetic studies may show that additional mycobacterial factors and metabolic pathways contribute to the host viability. 

### 4.2. Mechanistic Insights into Pyroptosis Induction by fdr Mutants

Several experiments confirmed that the *fdr* mutants induced robust pyroptosis of THP-1 cells (Figure 2). Gasdermin D (GSDMD) mediates pyroptosis by binding to the host membrane to form pores, which leads to the release of proinflammatory mediators, majorly IL-1β [66,67]. In humans, caspase-1 activates GSDMD, and the *M. tuberculosis* protein EST12 has been reported to induce GSDMD-induced pyroptosis in MΦ [45]. The induction of pyroptosis requires two inducing signals: (1) the expression of pro-IL-1β in response to TLR stimulation or MyD88 signaling and (2) the activation of inflammasome and caspase-1, requiring recognition of an intracellular danger signal by an NLR, such as NALP3. The *fdr* mutants clearly modulate signal 2 since *fdr*-induced MΦ cell death was dependent on caspase-1 and the NALP3 inflammasome (Figure 2) and resulted in IL-1β secretion (Figure 3A). Although we did not analyze GSDMD, the detection of IL-1β secretion in our experiment results strongly indicates its involvement in *fdr8*-induced pyroptosis. In addition, the observed *fdr8*-induced increase in MΦ IL-1β expression suggests that signal 1 is also modulated by *fdr* (Figure 4). *fdr*-induced pyroptosis was partially dependent on ROS, which is a confirmed activator of the NALP3 inflammasome [49,50], including during microbial infection [68]. The pyroptosis induced by *M. kansasii* has been attributed to the NALP3 inflammasome and ROS [41]. The data presented here provide further evidence that the NALP3 inflammasome has a role in the control of virulent *Mycobacterium* infection.

### 4.3. The Effects of Mycobacterial Suppression of Pyroptosis on Host Immunity

*fdr* inactivation had multiple immunological effects on MΦ and in vivo. Infection of THP-1 cells with *fdr* mutants resulted in increased IL-1β, TNF-α, and IL-4 (Figure 3). IL-1 and TNF-α are known to be critical for the control of tuberculosis [69,70]. Interestingly, IL-4 is also important for the control of inflammation during tuberculosis, protecting against the lung pathology concurrent with an unchecked inflammatory response [71]. Surprisingly, *fdr*-infected THP-1 cells showed significant secretion of IFN-γ, a cytokine normally secreted by T cells. Reports show that human MΦ can secrete IFN-γ in the presence of pro-inflammatory cytokines IL-12 and IL-18 [72,73]. While IL-12 showed only a trend for increase, IL-12 had an early peak and was endocytosed [74]. IL-18 was not tested as part of this assay but is processed by caspase-1 in a manner similar to IL-1β, and the secretion of the two are correlated [75]. Interestingly, IFN-γ has anti-inflammatory functions, including the promotion of caspase-1- and -3-dependent cell death and the inhibition of IL-1β, TNF-α, and IL-8 production [76]. Perhaps together, the IL-4 and IFN-γ seen in *fdr*-infected MΦ help modulate the pyroptotic inflammatory response. Finally, *fdr*-infected THP-1 cells secreted less IL-8 than wild-type-infected controls. This chemokine is a major mediator of the inflammatory response and recruits neutrophils. IL-8 secretion is normally repressed by the anti-inflammatory properties of IL-4 [77] and IFN-γ [69], consistent with the results of this study. Therefore, these data indicate that the pyroptosis induced by *fdr* strains results in a protective—and not pathological—immunological response in the host. 

This is supported by the growth of *fdr* mutants in MΦ and the immunogenicity of these strains in mice. The *fdr* mutants showed decreased growth in THP-1 cells. The induction of pyroptosis is associated with intracellular control of infection for several intracellular pathogens [42,78,79,80,81]. Moreover, in vivo, ELISPOT analysis confirmed that the inactivation of *fdr* genes and pyroptosis-suppressing mechanisms was beneficial to cellular immunity to *M. tuberculosis* in vivo. This group previously reported the first in vivo studies linking the subversion of mycobacterial MΦ apoptosis-suppressing mechanisms, by precise deletion of *secA2,* to the augmentation of *M. tuberculosis*-specific CD8 T-cell immunity [13]. Here, the inactivation of pyroptosis-suppressing *fdr* genes was robustly enhanced both CD4 and CD8 T-cell responses (Figure 3). The in vivo results are especially significant since a direct connection between pathogen-induced pyroptosis and adaptive immunity to *M. tuberculosis*-specific antigens have not been demonstrated. The induction of pyroptosis through non-microbial compounds has been demonstrated to augment CD4 and CD8 T-cell immunity [82,83], and the importance of pyroptosis to adaptive immunity to microbes has been predicted [25,55]. However, it was reported recently that Rv1759c, a secreted protein of *M. tuberculosis*-induced pyroptosis in MΦ by activating NLRP3 (NACHT, LRR, and PYD domains containing protein 3)–caspase-1/11–GSDMD–interleukin-1β (IL-1β) immune process but had no effect on T cells [45].Therefore, this work represents an important association of mycobacteria-induced NALP3-dependent pyroptosis with augmented cellular immunity. However, CFU burden in *fdr*-immunized mice showed no attenuation at this point (Appendix A). The ELISPOT results at 3 weeks post infection reflect a time point optimal to sampling of early cellular responses but represent a point prior to onset of active adaptive immunity, the main controlling factor of *Mycobacterium* burden in vivo [84]. Chai et al. (2022) reported that the protein tyrosine phosphatase B (PtpB, also called Rv0153c) from *M. tuberculosis* inhibits GSDMD-dependent pyropotosis by altering the phospholipid composition of the host membrane [27].This interaction has also been shown to be important in evading host immunity in a mouse model. Together, these data confirm that the suppression of pyroptosis is a potent immune evasion tactic of virulent *Mycobacterium*. 

### 4.4. The Role of fdr8 in Bacterial Homeostasis and Redox Balance

*fdr8* (BCG3787 or Rv3727) is annotated to encode a predicted oxidoreductase, similar to phytoene dehydrogenase *crtI*, which is involved in microbial carotenoid biosynthesis. Interestingly, carotenoids are protective against reactive oxygen species in bacteria species [85,86,87]. *ctrI* and/or carotenoid production is essential for virulence of *Rhodococcus equi* [88], *Staphylococcus aureus* [89], *M. marinum* [65,90], and *M. tuberculosis*. Interestingly, *M. marinum crtI* is required for growth and survival in MΦ [65,90]. Similarly, *M. tuberculosis* Rv3727 is required for survival in non-human primates [65]. Therefore, it is clear that Rv3727 plays an important role during host infection, although the functional data are currently lacking. 

Transcriptomic studies have provided valuable clues to the role of Rv3727 in *Mycobacterium* pathogenesis. Microarray experiments identified an over-representation of redox stress genes in an Rv3727-null mutant during MΦ infection (Figure 4A, left). Consistent with this, under-represented genes (Figure 4A, right) included genes involved in oxidation reactions or iron metabolism. These data suggest the *fdr8* mutant experiences greater redox stress in MΦ compared to wild-type BCG. Exposure of *fdr8*-null mutants to H_2_O_2_ stress, a classical ROS generator, did not negatively affect bacterial viability (Figure 5C,D). However, exposure to reductive stress, DTT, did affect growth (Figure 5A,B). Reductive stress is also linked to the production of ROS, as reduction of electron carriers can promote the formation of toxic ROS in the presence of oxygen [57,58]. The data suggest that BCG3787/Rv3727 is important for redox balance during reductive stress. 

In vivo studies show that *M. tuberculosis* experiences severe reductive stress in vivo [91], presumably due to the β-oxidation of host lipids. β-oxidation is associated with redox stress, free Fe^3+^, and ROS [58]. Lipid biosynthesis (polyketides and triacyglycerol) has been hypothesized to be an important reductive sink in *Mycobacterium* used to maintain redox balance when utilizing host cholesterol and lipids [92]. Consistent with this, several genes involved in fatty acid and protein biosynthesis were over-represented in an *fdr8*-null during MΦ infection, two processes that maintain bacterial redox balance and decrease reductive stress [57,92]. Moreover, *fdr8*-null exhibited a growth defect when grown in vitro on odd-chain lipids, a reductive-stress-inducing condition. Unlike even-chain fatty acids, odd-chain fatty acid catabolism produces propionyl-CoA units that must be processed, mainly through the methylcitrate cycle [93,94]. This feeds into the reductive arm of the TCA cycle (succinate to malate), unlike even-chain fatty acid catabolism, which yields acetyl-CoA and feeds into the glyoxylate bypass. In support of this, an increase in cellular propionyl-CoA during host lipid (cholesterol) catabolism has been reported for *M. tuberculosis* [95]. Perhaps an increase in cellular reducing equivalents induces reductive stress and ROS in the *fdr8* mutant during intracellular infection that cannot be overcome (Figure 6).

In addition to functioning as an antioxidant, mycobacterial carotenoid biosynthesis may serve as a valuable lipid reductive sink, similar to the biosynthesis of related terpenoids (polyketides) [57]. Carotenoids may also aid in the maintenance of lipid reductive sinks in mycobacteria and other bacteria. Carotenoid production is tied with the oxidation of NADPH [96,97] and the production of lipids such as triacylglyerols [98,99]. BCG3787/Rv3727 may contribute in this way to *M. tuberculosis* redox balance while utilizing host carbon sources (Figure 6). 

### 4.5. Mechanistic Links between fdr and the Induction of Pyroptosis

ROS is an inducing signal of the NALP3 inflammasome and pyroptosis in eukaryotic cells [49,50]. The transcriptome of *fdr8*-infected THP-1 cells did not identify an over-representation of host antioxidant functions (Appendix A). Therefore, the pyroptosis inducing ROS, as observed during *fdr* infection, is likely to be bacteria-derived (Figure 6). This is supported by mechanistic work characterizing the *fdr8* mutant (Figure 4 , Figure 5 and Appendix A), which indicates that BCG3787/Rv3727 is required for the balance of bacterial redox and ROS in the intracellular lifestyle of *Mycobacterium*. 

A recent report has linked NALP3 inflammasome activation with fatty acids and fatty acid–CoA derivatives in MΦ [100], and dependent on palmitate, ROS, TLR4 stimulation, and TNF-α, Mycobacteria stimulate TLR4 signaling [101], and TNF-α was seen up-regulated in *fdr*-mutant-infected THP-1 cells (Figure 3B). There is also ample proof that *Mycobacterium* changes its protein/enzyme profile and lipid profile in response to infection and redox stress [3,7,102,103]. Therefore, *fdr*-deletion-induced redox imbalance in the host may modulate fatty acid biosynthesis in such a way that the NALP3 inflammasome is induced in response to mycobacterial lipid. Future investigations will examine the NALP3-inducing signal modulated by *fdr* genes and whether the *fdr* strains exhibit differences in lipid profile during host infection or while utilizing host carbon sources.

## 5. Conclusions

In summary, our studies have identified novel mycobacterial pyroptosis-suppressing mechanisms that underscore the significance of metabolic and redox balance to the *M. tuberculosis* intracellular lifestyle and provide striking evidence that the metabolic state of *Mycobacterium* during infection is critical to the pathogen–host interaction. Therefore, central metabolic homeostasis constitutes a bona fide but perhaps underappreciate*,* host evasion virulence mechanism in virulent *Mycobacterium.* Importantly, this work presents a critical link between pathogen-induced pyroptosis of innate cells and augmented cellular immunity. Continued investigation of the requirements for *Mycobacterium* pathogenicity, including metabolic requirements, will no doubt prove invaluable to our evolutionary understanding of this complex pathogen and to the future rational design of drugs and vaccines to combat tuberculosis worldwide. 

## Figures and Tables

**Figure 1 pathogens-12-01109-f001:**
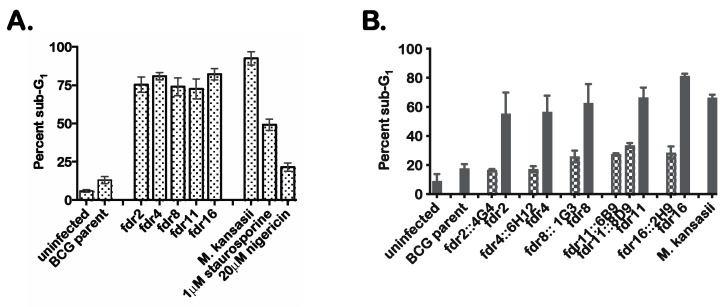
*fdr* mutants induce cell death of infected MΦ. (**A**). Secondary screening of candidate *fdr* mutants by sub-G_1_ analysis. Flow cytometric cell cycle analysis of infected THP-1 cells that were ethanol-fixed and stained with propidium idodide. Percent sub-G_1_ was calculated from the total gated population for each sample from three independent experiments. Staurosporine and nigericin treatments were for 4 h at 37 °C. (**B**) THP-1 cells were infected with mycobacteria strains at an MOI of 5:1, and 3 days post infection, infected cells were analyzed by sub-G1 analysis for cell death. Checkered bars = cosmid integrants; solid bars = *fdr* strains. Data are from three independent experiments.

**Figure 2 pathogens-12-01109-f002:**
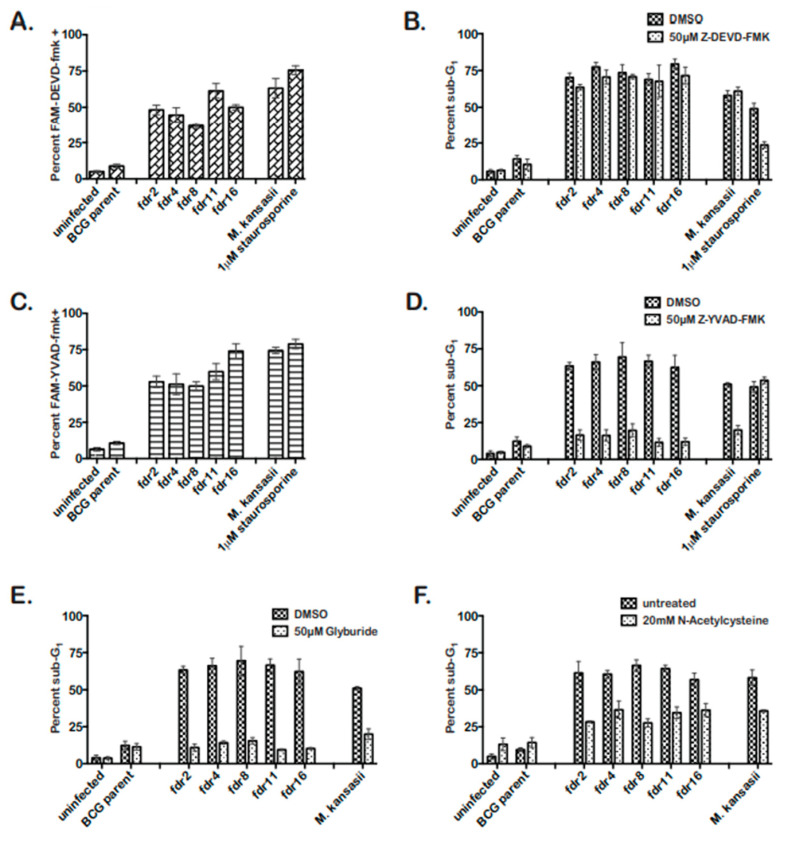
*fdr* mutants induce pyroptosis of infected MΦ, dependent on the NALP3 inflammasome and ROS. THP-1 cells were infected at an MOI 5:1 with mycobacteria strains. Three days post infection, cells were analyzed. (**A**) Caspase-3/7 was activated during *fdr* infection of THP-1 cells. Infected MΦ were stained for active caspase-3/7 staining and quantitated by microscopy. Data are from two independent experiments. (**B**) Caspase-3/7 inhibitor had no effect on *fdr*-induced cell death. THP-1 cells were infected and incubated in media containing 50 µM Z-DEVD-fmk until cell death analyses. Data are from three independent experiments. (**C**) Caspase-1 is activated during *fdr* infection of THP-1 cells. Infected MΦ were stained for active caspase-3/7 staining and quantitated by microscopy. Data are from two independent experiments. (**D**) Caspase-1 inhibitor abolished *fdr*-induced cell death. THP-1 cells were infected and incubated in media containing 50 µM Z-YVAD-fmk until cell death analyses. Data are from three independent experiments. (**E**) THP-1 cells were infected and incubated in media containing 50 µM glybyride until cell death analyses. Data are from three independent experiments. For all *fdr* infections, (**F**) ROS contributes to the induction of pyroptosis in *fdr*-infected THP-1 cells. THP-1 cells were infected and incubated in media containing 20 mM NAC until cell death analyses. Data are from three independent experiments.

**Figure 3 pathogens-12-01109-f003:**
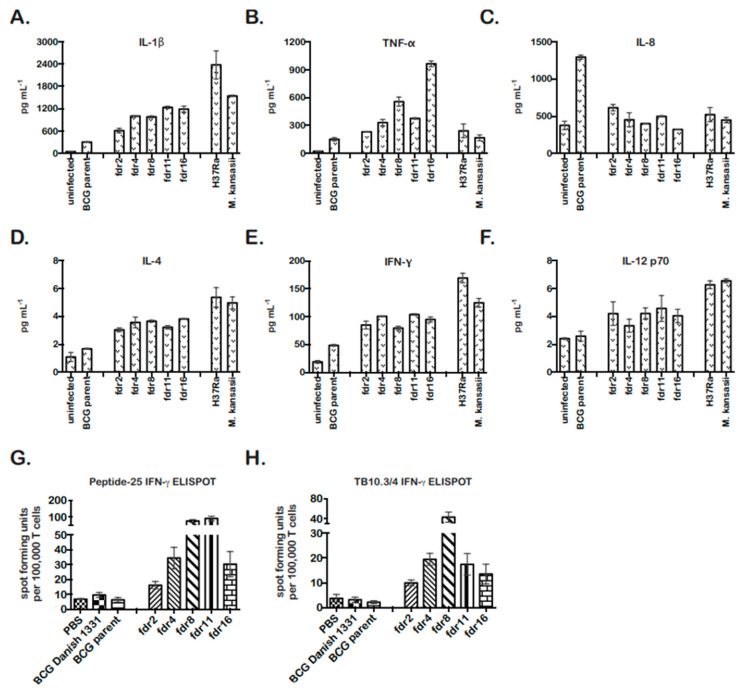
*fdr* mutants are more immunogenic in MΦ and in vivo. MSD MULTIARRAY analyses of mycobacteria-infected THP-1 supernatants for (**A**) IL-1β, (**B**) TNF-α, (**C**) IL-8, (**D**) IL-4, (**E**) IFN-γ, and (**F**) IL-12. Supernatants were collected on day 4 for analyses. Data are from three independent infections; means and standard deviations are shown. (**A**–**E**) For *fdr* infections, *p* < 0.05, relative to BCG parent control. (**G**) *fdr* mutants enhanced *M. tuberculosis*-specific CD4 T-cell responses in vivo. C57BL/6 mice were subcutaneously immunized with *fdr* mutants or controls, and at 3 weeks post immunization, spleens were harvested and assayed for *M. tuberculosis*-specific CD4 responses (peptide-25) by IFN-γ ELISPOT. (**H**) *fdr* mutants enhanced *M. tuberculosis*-specific CD8 T-cell responses in vivo. C57BL/6 mice were subcutaneously immunized with *fdr* mutants or controls, and at 3 weeks post immunization, spleens were harvested and assayed for *M. tuberculosis*-specific CD8 responses (TB10.3/.4) by IFN-γ ELISPOT; 3 mice per group. (**G**,**H**) For *fdr* infections, *p* < 0.05, relative to BCG parent control.

**Figure 4 pathogens-12-01109-f004:**
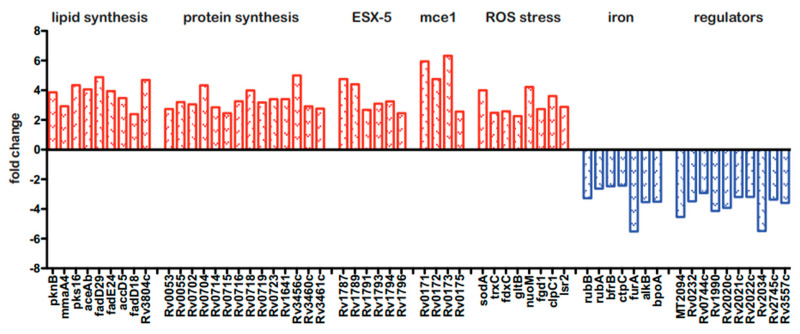
Bacterial and host transcriptional responses during *fdr8* MΦ infections. THP-1 MΦ were infected at an MOI of 5:1, and 3 days post infection, RNA was isolated for use in microarray analyses. Subset of bacterial genes differentially expressed in the intracellular environment in *fdr8* compared to wild-type bacterium. Fold change is given for each gene. Genes over-represented in *fdr8* compared to wild type are indicated in red bars, while genes under-represented are in blue bars. All microarray results were analyzed using TIGR MeV software (JCVI).

**Figure 5 pathogens-12-01109-f005:**
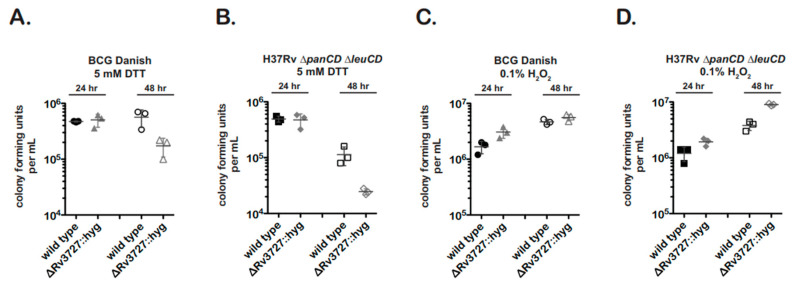
An *fdr8*-null mutant exhibits a growth defect when grown under reductive stress conditions. First, 3 × 10^6^ log phase bacteria were resuspended in media containing 5 mM DTT (**A**,**B**) or 0.1% H_2_O_2_ (**C**,**D**). At 24 h and 48 h timepoints, cultures were serial diluted 10^0^–10^−7^ and 5 µL spotted for cfu. Results are representative of two independent experiments with similar results.

**Figure 6 pathogens-12-01109-f006:**
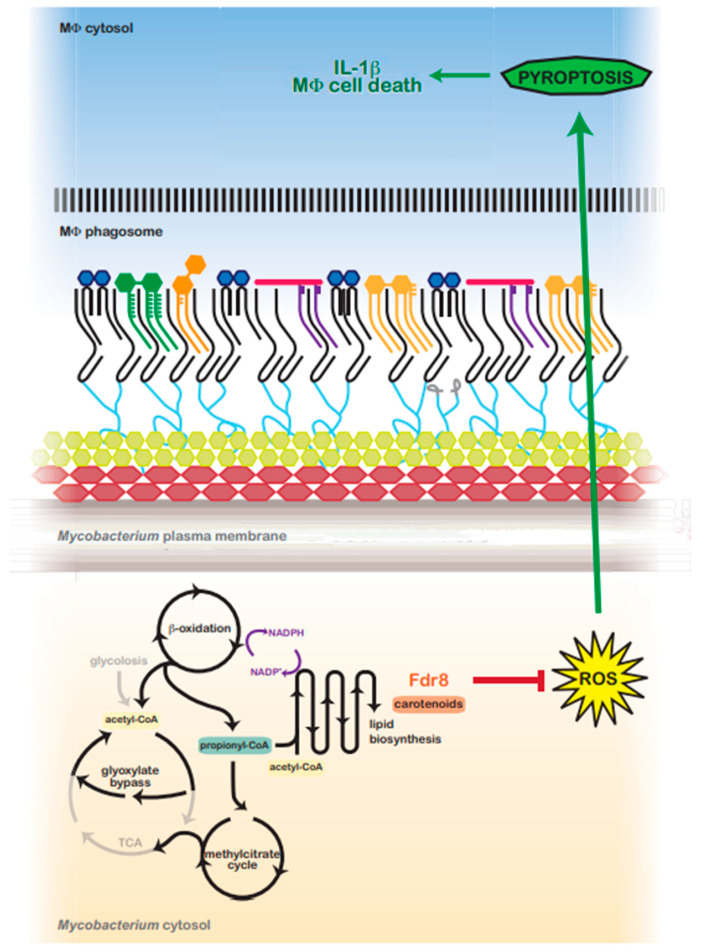
Predicted model for *fdr8*-regulated redox balance and MΦ pyroptosis suppression by intracellular *Mycobacterium*. In the host, *Mycobacterium* utilizes host lipids (cholesterol) as a primary carbon source. β-oxidation catabolizes even and odd-chain lipids into acetyl-CoA or propionyl-CoA units, respectively, generating NADPH and a reductive environment that must be balanced. Acetyl-CoA is metabolized via the glyoxylate shunt, bypassing the reductive arm of the TCA cycle. However, propionyl-CoA is primarily metabolized via the methylcitrate cycle, which feeds into the TCA cycle, generating reduced cofactors and contributing to reductive stress. To balance reductive stress, *Mycobacterium* utilizes several metabolic processes as reductant sinks. Lipid anabolism functions as a main means to regenerate NADP^+^. *fdr8* and carotenoid biosynthesis may contribute directly to this process, or carotenoids may protect against reductive-stress-induced ROS. ROS production is an inducing signal for the NALP3 inflammasome, and in the absence of *fdr8* and/or other *fdr* genes, ROS is increased, resulting in the induction of pyroptosis, IL-1β secretion, and macrophage cell death.

**Table 1 pathogens-12-01109-t001:** Functional annotations of identified *fdr* genes.

*fdr*	BCG ORF	Rv Ortholog	Gene Name	Predicted Function ^*a*^
2	BCG1866	Rv1831		Hypothetical protein
4	BCG3824c	Rv3765c	*tcrX*	Probable 2-component RR
8	BCG3787	Rv3727		Probable phytoene desaturase *crtI*
11	BCG1264c	Rv1204c		Hypothetical protein
15	BCG2813c	Rv2795c		Hypothetical protein
	BCG3787	Rv3727		Probable phytoene desaturase *crtI*
16	BCG1058	Rv1001	*arcA*	Probable arginine deiminase

*^a^* Annotated functions provided by TubercuList and BCGList.

**Table 2 pathogens-12-01109-t002:** Growth rate of *Mycobacterium* strains, calculated from OD_600nm_ data by Logistic non-linear regression curve fit *^a,b^*.

	Growth Rate (h^−1^) *^c^*						
Strain	Glu	Glyc	Acet	Prop	But	Val	Cap
BCG-Danish	0.031	0.022	0.03	0.017	0.042	0.051	0.022
BCG Danish ∆BCG3787::*hyg*	0.013	0.016	0.033	**0.008**	0.037	**0.003**	0.021
mc^2^6206	0.008	0.016	0.015	0.01	0.058	0.02	0.022
mc^2^6206 ∆Rv3727::*hyg*	0.008	0.018	0.013	**0.002**	0.039	**0.004**	0.016

*^a^* Graphpad Prism 5 software used for non-linear regression curve fit analysis, using the Logistic formula y = y_max_ ∗ y_0_/[(y_max_ − y_0_) ∗ exp(−k ∗ x) + y_0_)]. *^b^* The mean r^2^ correlation value for curve fitting was 0.9836 ± 0.021. *^c^* Growth rate = k = ln 2/doubling time. The typical growth rate for a *Mycobacterium* strain with a 24 h doubling time is 0.029 h^−1^.

## Data Availability

Data available as Appendix A.

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
