# Peer review of "Mycobacterium tuberculosis Central Metabolism Is Key Regulator of Macrophage Pyroptosis and Host Immunity"

_pathogens, 2023, doi:10.3390/pathogens12091109_

Round 1

Reviewer 1 Report

See attached

Author Response

An itemized list of the changes and additions made is presented below (in red), along with reviewers’ original comments (in black) for reference. Additionally, we have made the changes in the word file provided using track changes, so the reviewers review the text.

Reviewer 1

We thank the reviewer for their keen eye. All typographical errors listed below have been corrected, as suggested by the reviewer:

L17 – Spacing in front of Analysis needs correcting.

L53 – space after [1,2] needs removing.
L58 – space after [8-12] needs removing.
L62 – space after [13-23] needs removing.

L66 – space after [28] needs removing.
L73 – space after “immunity.” needs correcting.
L93 – °C not ° C (needs correcting throughout the manuscript).
L98 – adapted from .......
L168 – 4 mg
L231 – space after [17,29] needs removing.
L232 – space after [30,31] needs removing.
L234 – space after [32-34] needs removing.
L241 – can the 17 candidate mutants be presented in a supplementary table?
L244 – space after [35] needs removing.

L278 – change to “3.2. Mechanism
L286 – space after [23,36] needs removing.
L291 – space after [37] needs removing.
L354 – space after [51] needs removing.
L365 – space after IL-18 needs removing.
L371 – 5 mutants?
Figure 3 legend has many typographical errors, spaces, and additional period points.
L392 – space after [52-54] needs removing.
L396 – space after “immunity.” needs correcting.
L398 – space after “ELISPOT.” needs correcting.

L412 – space after “analysis.” needs correcting. L423 – error in the title, symbol missing.
L438 – space required after Also,
L456 – TABLE change to Table

L457 – TABLE change to Table
L462 – remove . after expression, add space after , (very long sentence which may need to be edited). L493 – TABLE change to Table
L503 – TABLE change to Table
L510 – Move Table 2 heading to above the table
L526 – TABLE change to Table
L528 – space after [5,63] needs removing.
L529 – space after [63], remove space after MF. Merge sentences to 531.
L548 – space after [49,50] needs removing.
L565 – space after [76] needs removing.
L569 – irregular spacing after [77], space after [69] needs removing.
L575 – space after -81] needs removing.
L580 – space needed after [13].
L589 – space needed after [45].
L595 – space after [84] needs removing.
L597 – space needed after [27].
L605 – space after -87] needs removing.
L606 – spacing incorrect
L608 – space after [65] needs removing.
L629 – space after ,92] needs removing.
L642 – space needed after ,97] and removed after 99]
L654 – Title error
L656 – beta missing

L152 – mL-1?

The usage of μg mL-1 is a convection used to present μg/mL without the use of a slash. It can also be represented as μg.mL-1. To avoid confusion, we have changed all instances in the text to μg.mL-1

Figure 1 – can the scales used in the two figures be altered to be the same, fdr8 and 15 are alternatively used in this figure, the text does support this but the data for both could have been included in this figure and the subsequent choice to only use fdr8 for all subsequent experiments is justified.

This was an error on our part. The fdr8 has been placed in figure 1B to avoid confusion. The text still references fdr15, and we have left the justification for its exclusion from analyses on (Line 295).

L372 – The growth observed under nutritious growth conditions i.e., 7H9 showed no impairment, which would be expected but how would these strains perform in a more minimal media? This might change the conclusion made by the authors L374-376 if a growth defect was observed.

The reviewer raises a valid point, as fdr8 has a growth defect when grown in medium containing odd-chain lipids as a carbon source. As this screen identified several pathways involved in bacterial metabolism, it is possible that several of the mutants will have growth defects in different minimal media formulations. However, full analysis of the growth of all strains is outside the scope of this manuscript. We have expanded the discussion to discuss the growth of the mutants in vitro versus intracelluarly (see line 593).

Reviewer 2 Report

In this research article, the authors investigated the impact of bacterial loci on pyroptosis, a form of cell death, and have shown that the induction of pyroptosis by Mycobacterium improves the immune response to M. tuberculosis in terms of quality and quantity. The manuscript is well-written, and the experimental design and data analysis are robust. I would recommend the following comments to the authors.

Minor Comments

Point 1: This study is highly unique and original; there are a few limitations: This study was conducted using M. bovis BCG, a vaccine strain of M. tuberculosis, and the results may not be directly applicable to other strains of M. tuberculosis.

Point 2: This study focuses on the role of metabolic genes in the induction of cell death in infected macrophages and does not investigate other factors that may contribute to pathogen virulence and host immunity.

Point 3: The study did not investigate the potential role of other bacterial metabolic pathways in regulating macrophage pyroptosis and host immunity.

Point 3: The authors need to add one figure describing the fundamental experimental workflow and immunization schedules. Otherwise, it would be difficult for the reader to capture the overall picture of the study. Overall, I could not fault the experiments or the interpretation. Future experiments in memory response would be informative.

Point 4: A few minor grammar mistakes need to be corrected thoroughly.

Good Luck!

A few minor grammar mistakes need to be corrected thoroughly.

Author Response

An itemized list of the changes and additions made is presented below (in red), along with reviewers’ original comments (in black) for reference. Additionally, we have made the changes in the word file provided using track changes, so the reviewers review the text.

Point 1: This study is highly unique and original; there are a few limitations: This study was conducted using M. bovis BCG, a vaccine strain of M. tuberculosis, and the results may not be directly applicable to other strains of M. tuberculosis.

We agree with the reviewer, however such a screen in virulent tuberculosis would not have not been feasible. However, BCG and M. tuberculosis are genetically identical, and in behave similarly in macrophages in the time frame of our screen (3 days). This is discussed on line 252 of the text. Nonetheless, as least for fdr8, we agree with the reviewer and therefore reconstructed the deletions in auxotrophic strain of M. tuberculosis (mc26206) to recapitulate the pyroptosis phenotype in THP1 cells and primary macrophages. This data is included in Figure S5. As the focus of the second half of the manuscript is on the characterization of fdr8, studying the remaining mutations in M. tuberculosis is currently outside the scope of this manuscript.

Point 2: This study focuses on the role of metabolic genes in the induction of cell death in infected macrophages and does not investigate other factors that may contribute to pathogen virulence and host immunity.

We agree with the reviewer, the non-biased screen was designed to identify mutant strains that induced cell death in macrophage therefore several bacterial pathways could feasibly be involved. Surprisingly, the majority of genes identified were genes in bacterial metabolism. It is unknown why these were our main hits, but likely implies that the intracellular metabolism of the bacterium is more important to host viability than one would assume. The implications and mechanisms of this will be explored in future studies. However, we have added text to our discussion (line 605) to raise this point.

Point 3: The study did not investigate the potential role of other bacterial metabolic pathways in regulating macrophage pyroptosis and host immunity.

Yes, as mentioned above we employed and analysed a genetic screen, which is a powerful tool to identify major pathways that modulate macrophage viability during mycobacterium infection. It is a limitation of all genetic screens that modest phenotypes may fall below the screen threshold. Therefore other bacterial factors may be involved in macrophage cell death modulation. However, the direct study of selected bacterial metabolic pathways that affect macrophage pyroptosis and host immunity is outside the scope of this study. However, we have added text to our discussion (line 605) to raise this point.

Point 3: The authors need to add one figure describing the fundamental experimental workflow and immunization schedules. Otherwise, it would be difficult for the reader to capture the overall picture of the study. Overall, I could not fault the experiments or the interpretation. Future experiments in memory response would be informative.

We respectfully disagree with this minor comment. Since the immunization experiment is only part of the entire manuscript, we feel that the methods section (line 160) provides all details necessary to repeat the experiment. Additionally, this protocol is standard in the M. tuberculosis field.

Point 4: A few minor grammar mistakes need to be corrected thoroughly.

These have been addressed.